# Construction of long non-coding RNA- and microRNA-mediated competing endogenous RNA networks in alcohol-related esophageal cancer

Quan Du[ⵔ], Ren-Dong Xiao[ⵔ], Rong-Gang Luo, Jin-Bao Xie, Zu-Dong Su*, Yu Wang[iD]*

Department of Thoracic Surgery, First Affiliated Hospital of Fujian Medical University, Fuzhou, People's Republic of China

ⵔ These authors contributed equally to this work.
* wyfz1968@163.com (YW); Szd1245@163.com (ZDS)

## Abstract

The current study aimed to explore the lncRNA–miRNA–mRNA networks associated with alcohol-related esophageal cancer (EC). RNA-sequencing and clinical data were downloaded from The Cancer Genome Atlas and the differentially expressed genes (DEGs), long non-coding RNAs (lncRNAs, DELs), and miRNAs (DEMs) in patients with alcohol-related and non-alcohol-related EC were identified. Prognostic RNAs were identified by performing Kaplan–Meier survival analyses. Weighted gene co-expression network analysis was employed to build the gene modules. The lncRNA–miRNA–mRNA competing endogenous RNA (ceRNA) networks were constructed based on our *in silico* analyses using data from miRcode, starBase, and miRTarBase databases. Functional enrichment analysis was performed for the genes in the identified ceRNA networks. A total of 906 DEGs, 40 DELs, and 52 DEMs were identified. There were eight lncRNAs and miRNAs each, including *ST7-AS2* and miR-1269, which were significantly associated with the survival rate of patients with EC. Of the seven gene modules, the blue and turquoise modules were closely related to disease progression; the genes in this module were selected to construct the ceRNA networks. SNHG12–miR-1–ST6GAL1, SNHG3–miR-1–ST6GAL1, SPAG5-AS1–miR-133a–ST6GAL1, and SNHG12–hsa-miR-33a–ST6GA interactions, associated with the N-glycan biosynthesis pathway, may have key roles in alcohol-related EC. Thus, the identified biomarkers provide a novel insight into the molecular mechanism of alcohol-related EC.

## Introduction

Esophageal cancer (EC), a cancer that usually begins in the cells lining the inner layer of the esophagus, is one of the most commonly diagnosed cancer and a leading cause of cancer-associated mortality [1]. The National Cancer Institute estimated approximately 17,290 new cases and 15,850 deaths from EC in the United States in 2018 [2]. Typically, EC remains asymptomatic until a relatively late stage, by when it metastasizes to tissues or lymph nodes surrounding

**Data Availability Statement:** All relevant data are within the manuscript and its Supporting Information files.

**Funding:** This work was financially supported by the Startup Fund for Scientific Research of Fujian Medical University (No. 2017XQ1097),The funders had no role in study design, data collection and analysis, decision to publish, or preparation of the manuscript.

**Competing interests:** The authors have declared that no competing interests exist.

the esophagus [3]. This poses a significant challenge in the treatment of EC and may be responsible for the low survival rate. It is reported that EC patients have a very poor prognosis with the 5-year survival ranging from 5 to 43% based on the stage and an overall 5-year survival of 19% [4].

Many factors, such as age, sex, alcohol consumption, and obesity, can be attributed to the development of EC, [5]. Alcohol is a chief risk factor for EC because of its worldwide prevalence and high carcinogenicity [6]. The mechanisms of ethanol-induced carcinogenesis are intensively associated with its metabolism [6]. In the liver, ethanol is converted to acetaldehyde by the enzyme alcohol dehydrogenase (ADH) and subsequently to acetate by another enzyme aldehyde dehydrogenase-2 (ALDH2). Ethanol is not only associated with increased cell membrane permeability, but also promotes carcinogen penetration into the mucosal epithelial cells as it acts as a solvent [7]. Ethanol itself is not a direct carcinogen, but shows its carcinogenic effects via downstream metabolic products that function as a co-carcinogen and/or tumor promoter [6, 7]. Moreover, alcohol consumption has been found to contribute to increased cell proliferation in the esophageal mucosa of rats [8]. Acetaldehyde, a product of ethanol metabolism, is concentrated in the upper digestive tract through alcohol oxidation via microbes, parotid glands, and mucosal cells [9]. High acetaldehyde concentration induces hyperplasia in the epithelia of upper digestive tract in rats [8, 10]. Accordingly, we hypothesized that acetaldehyde accumulation during ethanol metabolism may be involved in EC progression. In fact, certain epidemiological studies have clearly demonstrated that inactive ALDH2 encoded by *ALDH2\*1/2\*2* genotype, which induces an increased acetaldehyde accumulation after alcohol consumption, is a major risk factor for upper aero-digestive tract cancer development, particularly EC [11, 12].

The other factor that may contribute to EC development is the formation of reactive oxygen species (ROS) during ethanol oxidation [13]. Elevated rates of ROS production have been observed in almost all cancers, where they promote many aspects of tumor development and progression [14]. Besides, folate metabolism disruption has been hypothesized to be associated with alcohol-mediated carcinogenesis [15]. ROS accumulation and folate deficiency can induce genetic alterations that are strongly implicated in carcinogenesis [16]. ROS accumulation can increase the levels of monocyte chemotactic protein-1 and vascular endothelial growth factor [17], two key mediators of tumor angiogenesis and metastasis. In addition, the levels of two metalloproteinases, MMP2 and MMP9, also increase, which can induce extracellular matrix breakdown and cell motility, thus favoring tumor metastasis [18]. Folate deficiency has been observed to alter DNA methylation and disrupt DNA integrity and repair. Hence, it can also lead to altered expression of some critical tumor related genes such as *p53*, thereby enhancing carcinogenesis [19]. However, the molecular mechanism underlying the association of alcohol consumption and EC remains ambiguous.

The aberrant expression of protein-coding mRNAs and non-coding RNAs, as well as the their regulatory networks, has been closely associated with cancer initiation and development [20]. The non-coding RNAs comprise small non-coding RNAs, long non-coding RNAs (lncRNAs), and very long non-coding RNAs. miRNAs are short and endogenously expressed non-coding RNAs that can sequence-specifically bind to their target genes and negatively control their expression via silencing translation and/or catalyzing mRNA destabilization [21]. lncRNAs contain the miRNA-response elements that can serve as miRNA "sponges" and thereby compete with target mRNAs for binding to specific miRNAs serving as competing endogenous RNA (ceRNA) [21, 22]. Increasing number of studies has revealed the crucial roles of miRNAs/lncRNAs in the pathological progression of EC, including the proliferation, apoptosis, invasion, angiogenesis, metastasis, chemoradiotherapy resistance, as well as stemness of EC, indicating the potential clinical applications value of these non-coding RNAs as diagnostic and prognostic biomarkers [23–26]. Many computational models were then

developed to identify disease-related miRNAs and lncRNAs [27–29]. The lncRNA/miRNA/ mRNA regulatory axes have been reported to be involved in the progression of EC. The lncRNA HOX transcript antisense RNA (*HOTAIR*) can sponge *miR-1* and upregulate cyclin D1 (*CCND1*) expression, thus facilitating the tumorigenesis of esophageal squamous cell carcinoma (ESCC) [30]. However, studies focusing on the RNA regulatory networks involved in alcohol-related EC development are exceptionally rare.

In the current study, we downloaded the RNA profiles of patients with alcohol-related and non-alcohol-related EC from The Cancer Genome Atlas (TCGA) database (https://gdc-portal. nci.nih.gov/) to systematically investigate the lncRNA–miRNA–mRNA interactions associated with EC using multiple *in silico* methods. Identifying susceptible biomarkers, especially the lncRNA–miRNA–mRNA networks can facilitate a better understanding of the etiology of EC, thus providing improved and effective targets for the therapeutic management of the disease.

## Materials and methods

### Data acquisition

The RNA-sequencing fragments per kilobase of transcript per million fragments mapped (FPKM) data of mRNAs and miRNAs for EC specimens and corresponding clinical information were retrieved from TCGA database. The sequencing data for a total 198 mRNAs and miRNAs each obtained from IlluminaHiSeq_RNASeq and IlluminaHiSeq_miRNASeq sequencing platforms, respectively, were retrieved. The samples of mRNAs seq-data and the samples of miRNA seq-data were matched with the sequencing barcodes, and 190 samples were obtained after matching using sequencing barcodes, including 18 normal and 172 tumor samples. Among the 172 tumor samples, there were 120 alcohol-related EC tumor samples and 50 non-alcohol-related EC tumor samples according to the alcohol consumption status of the patients. The rest two samples had no records for alcohol consumption status. Finally, the 170 tumor samples were used in the following analysis.

### Data pretreatment and differential expression analysis

HUGO gene nomenclature committee (HGNC; http://www.Genenames.org/) is responsible for approving unique symbols and names for human loci, including protein-coding genes, non-coding RNA genes, and pseudogenes. A primary search was performed at the website of HGNC that led to the retrieval of a total of 19,004 protein-coding genes and 2,775 lncRNAs in humans. Bases on this information, the lncRNAs and mRNAs associated with esophageal carcinoma progression were retrieved from TCGA.

Low-abundance RNA with expression value less than one were filtered out. Differential analyses of RNAs between the alcohol-related and non-alcohol-related groups were conducted using the edgeR package (version 3.0.1) [31]. edgeR is a Bioconductor software package developed to analyze the replicated count-based expression data. An over-dispersed Poisson model was used to deal with both biological and technical variability. Empirical Bayes procedures were used to moderate the degree of over-dispersion across genes and improve the inference reliability. The differentially expressed genes (DEGs), miRNAs (DEMs), and lncRNAs (DELs) were identified upon satisfying the selection criteria including false discovery rate (FDR) < 0.05 and |fold change (FC)| > 1.5.

### Correlation of DEGs, DEMs, or DELs with clinical features

All the relevant clinical information was retrieved from the afore-mentioned databases or datasets. Based on the following dichotomous variables, the patients were divided into different

subgroups: age ($\geq$ 60 vs. < 60), clinical stage (III+IV vs. I+II), sex (female vs. male), pathological tumor-node-metastasis (TNM) stages (T3+T4 vs. T1+T2, N2+N3 vs. N0+N1, MX vs. M0), neoplasm histologic grade (G3+G4 vs. G1+G2), new tumor (Yes vs. No), and smoking status (Yes vs. No). The mRNAs, miRNAs, and lncRNAs, which were differentially expressed between any above paired subgroups, were identified using the edgeR package [31]. Similarly, FDR < 0.05 and |FC| > 1.5 were selected as the cutoff values.

## Selection of prognostic mRNAs, miRNAs, and lncRNAs

The survival information of each patient was extracted, based on which the survival analysis with a univariate cox model using the survival function in R [32] was performed on the selected DEGs, DEMs, and DELs between alcohol-related and non-alcohol-related samples to identify the prognostic RNAs. The median expression value of the potential prognostic miRNAs, lncRNAs, or mRNAs was used as the cutoff to classify the patients into upregulated and downregulated groups. The Kaplan–Meier (KM) survival analysis was used to assess the survival difference of the two groups and the significance was estimated by the log-rank test.

## Construction of the co-expression network

In the present study, the weighted gene co-expression network analysis (WGCNA) method, which is a systems biology method to describe the correlation patterns among genes across microarray samples, was used to construct the co-expression network [33]. In the co-expression network, the nodes and lines represented gene and relevance modules, respectively.

It was hypothesized that the network could satisfy the scale-free law [34] and it was constructed as follows:

A, Defining the correlation matrixes: the Pearson's correlation coefficient was used to calculate the similarity between any two genes, and the correlation matrix was formed as follows:

$$S_{mn} = |cor_{(m,n)}| \tag{1}$$

B, Defining the weighted adjacency matrixes: the exponential adjacency function in the WGCNA algorithm was used to measure the relation index of a gene pair, which is the exponential weighted $\beta$ square of the correlation coefficient. The adjacency function was calculated based on the following formula:

$$a_{mn} = power_{(S_{mn}, \beta)} \tag{2}$$

C, Determination of the weighted $\beta$: the weighted $\beta$ was determined based on the scale-free network law that the correlation coefficient of log (k) and log (p(k)) is at least 0.9 (k is the node connectivity and p is the probability).

D, Defining a measure of the node dissimilarity: After selecting the threshold parameter $\beta$, the correlation matrix $S_{mn}$ was switched to the adjacency matrix $a_{mn}$, and subsequently converted to the topological overlap matrix $\Omega$.

$$\Omega = w_{mn} \tag{3}$$

$$w_{mn} = \frac{l_{mn} + a_{mn}}{min\{k_m, k_n\} + 1 - a_{mn}} \tag{4}$$

In the formula (4), $l_{mn}$ represents the sum of the adjacency coefficients of the nodes connected to both m and n. $k_m$ is the sum of adjacency coefficients of the nodes only connected with m. In cases where there is no connection between genes m and n and no neighbors are

shared by them, $w_{mn} = 0$. The dissimilarity of nodes, the basis of network construction, was measured as $d_{mn} = 1-w_{mn}$.

E, Identifying gene modules: hierarchical clustering was performed according to the dissimilarity coefficient to assign genes with similar expression profiles into gene modules. The Dynamic Hybrid cut, a bottom-up algorithm that improves the detection of outlying members of each cluster, was used to identify the gene modules through two steps. First, the preliminary clusters were identified as branches when they satisfied the following criteria: (1) they included a certain minimum number of genes; (2) the genes too far from a cluster were excluded even if they belonged to the same branch of the dendrogram; (3) each cluster was distinct from its surroundings; and (4) the core of each cluster, defined as the tip of the branch, was tightly connected. Second, all previously unassigned objects were tested for sufficient proximity to preliminary clusters; if the nearest cluster was close enough, the object was assigned to that cluster.

## Prediction of lncRNA–miRNA interactions

The lncRNA–mRNA interactions from miRcode (miRcode 11, http://www.miRcode.org/) [35] and starBase (starBase V2.0, http://starbase.sysu.edu.cn/) [33] databases were extracted and integrated to predict the DEL-related lncRNA–miRNA interaction pairs. These interactions were then mapped to DEMs to retrieve the DEL–DEM interaction pairs.

## Prediction of target genes of miRNAs

The miRTarBase (http://miRtarbase.mbc.nctu.edu.tw) is a database available for the latest and extensive experimentally validated miRNA–target interaction information [36]. In the current study, we have used the miRTarBase database (release 6.0) to predict the potential target genes of the selected DEMs. The miRNA–target gene network was then constructed by integrating the DEG interactions in the protein–protein interaction network. The network was visualized with the help of Cytoscape software [37].

## Construction of ceRNA regulatory networks

The ceRNA networks, which are the lncRNA–miRNA–mRNA regulatory networks, were constructed by integrating the lncRNA–miRNA and miRNA–target gene interactions. As documented, topological centralities (degree, closeness, as well as betweenness) are wildly used to study the network topology properties. Herein, degree centrality, the simplest index, was used to acquire the key nodes by ranking the scores. The higher-score nodes were regarded as important nodes in the network, namely hubs. The Gene Ontology (GO) and Kyoto Encyclopedia of Genes and Genomes (KEGG) enrichment analyses of the candidate genes in this network was conducted using the Fisher's exact test to identify the functions and pathways closely related to EC. The detailed Fisher's exact test algorithm is shown as follows:

$$p = 1 - \sum_{i=0}^{x-1} \frac{\binom{M}{i}\binom{N-M}{K-i}}{\binom{N}{K}} \tag{5}$$

Where N, M, and K indicate the total gene counts in the whole genome, gene counts in the pathway, and counts of DEGs, respectively. The p value indicates the chance that at least "x" of "k" genes is enriched in a specific function or pathway term.

The flow chart of method is shown in Fig 1.

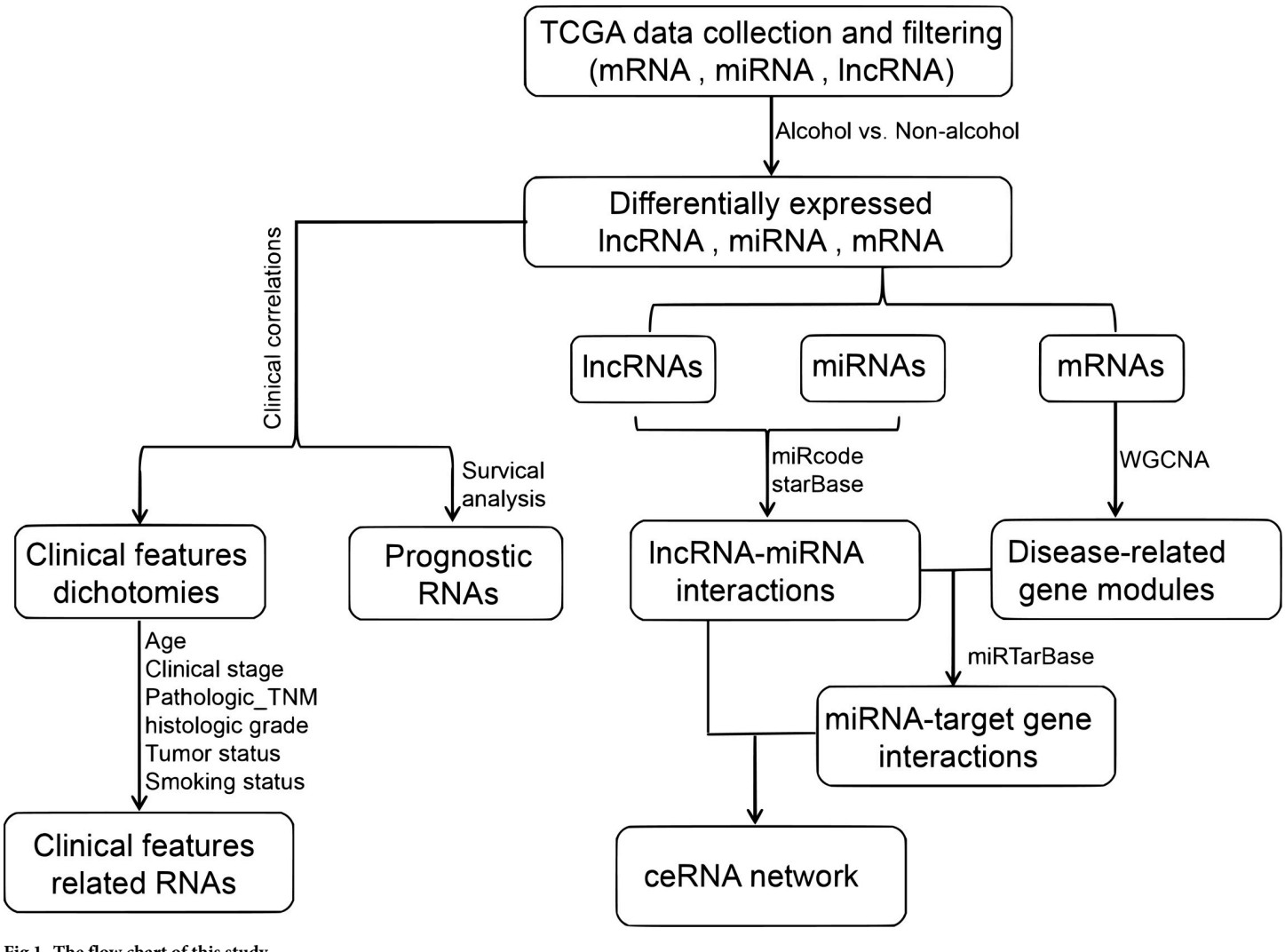

**Fig 1. The flow chart of this study.**

## Results

### Identification of EC-related lncRNAs, miRNAs, and mRNAs

Our *in silico* analyses of the RNA-seq data led to the identification of a total of 1,046 human miRNAs, 735 lncRNAs, and 17,580 protein-coding mRNAs. Post filtering out of the low abundant RNAs, 398 lncRNAs, 569 miRNAs, and 13,420 mRNAs were finally obtained. The filtration step apparently led to increased peaks of expression density for all three types of RNAs (Fig 2A). Remarkably, the lncRNA expression levels were lower than mRNA and miRNA expression levels.

The ensuing differential expression analysis led to the identification of a total of 906 DEGs, 40 DELs, and 52 DEMs. Interestingly, hierarchical clustering analyses of all three kinds of differentially expressed RNAs led to the generation of heat maps, which could efficiently distinguish patients with alcohol-related esophageal carcinoma from those with non-alcohol-related esophageal carcinoma (Fig 2B).

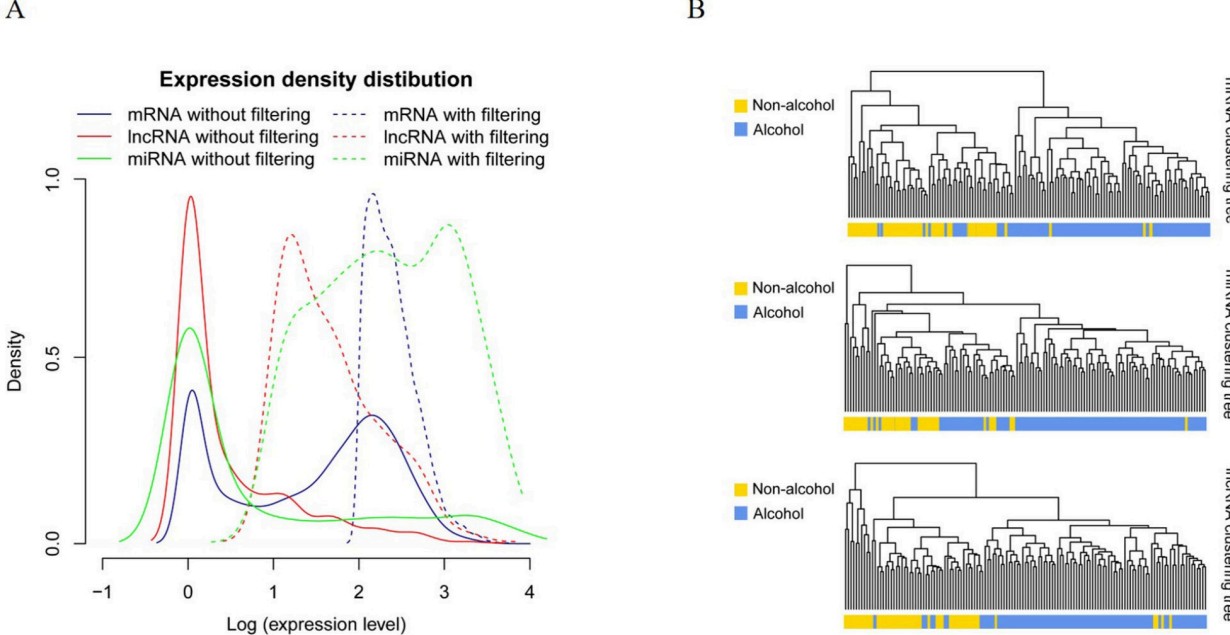

**Fig 2. Cluster analysis of the selected RNAs.** A. The distribution of expression density of mRNAs, miRNAs, and lncRNAs. The peak expression density increased after filtering the RNAs with expression level < 1. B. Clustering map of samples based on differential expression of genes, miRNAs, and lncRNAs. In which, the samples in one group tended to cluster together.

## Relationship between DEGs and clinical features in EC

To investigate whether the differentially expressed RNAs exhibit any association with EC clinical features, the cancer patients were divided into several subgroups according to the aforementioned eight dichotomous variables. Subsequently, relevant comparison analysis for RNA expression profiles was performed and the DEGs, DEMs, or DELs in each comparison groups were identified (S1–S3 Tables).

## Identification of prognostic RNAs in EC

To facilitate identification of RNAs with prognostic significance in EC, univariate cox regression analysis was performed on the differentially expressed RNAs (Table 1). Our analyses remarkably identified eight lncRNAs, including four upregulated (*C17orf100*, *RNU11*, *MOR-C2-AS1*, and *SNHG10*) and four downregulated ones (*ASMTL-AS1*, *ST7-AS2*, *MIR210HG*, and

**Table 1. The prognostic mRNAs, miRNAs and lncRNAs.**

| RNA | Up-regulated | Down-regulated |
|-----|--------------|----------------|
| lncRNA | C17orf100, RNU11, MORC2-AS1, SNHG10 | ASMTL-AS1, ST7-AS2, MIR210HG, AFAP1-AS1 |
| miRNA | hsa-mir-1269, hsa-mir-421, hsa-mir-340 | hsa-mir-1293, hsa-mir-135b, hsa-mir-299, hsa-mir-412, hsa-mir-627 |
| mRNA | FCHSD2, BMP6, EPB41L2, FAM111A, WDR54, DMAP1, CNN1, MGMT, PARVB, AGPAT5, GDF15, HMMR, PIM2, TIMM21, PRR15L, ZBTB2, RIPK2, LAGE3, FUNDC1, HOOK1, FILIP1L, SPCS2, CDK10, PGAP3 | CCDC51, KIF13A, ANO10, MAN2A1, PBRM1, ST6GALNAC6, RAB38, FAM83A, METRN, MAN2B2, ADAM17 |

hsa, Homo sapiens; miR, microRNA; lncRNAs, long non coding RNAs.

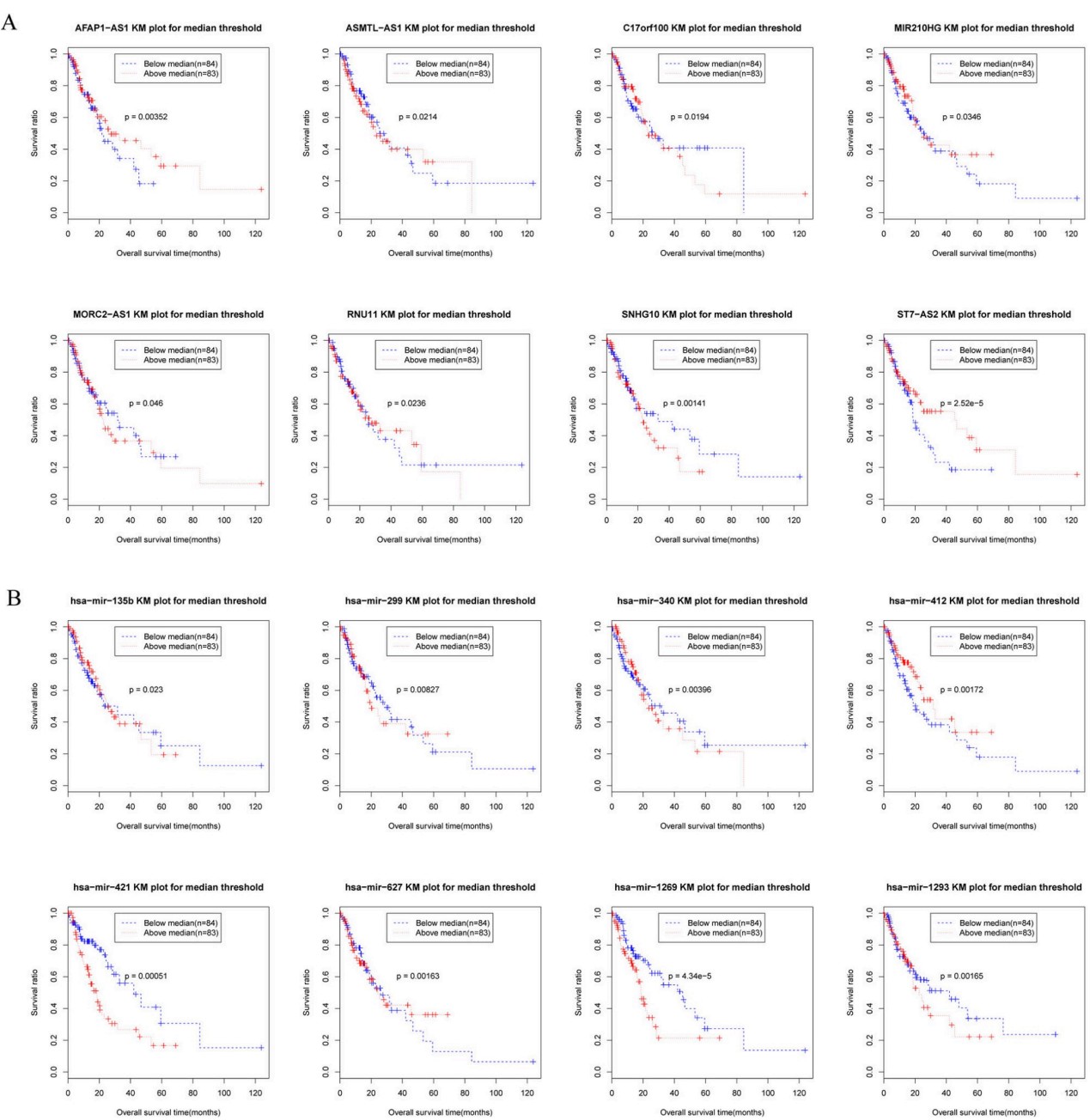

**Fig 3. Prognosis related lncRNAs and miRNAs.** Univariate Cox regression for differentially expressed lncRNAs and miRNAs identified eight lncRNAs and eight miRNAs were associated with overall survival of patients with alcohol-related esophageal cancer. Kaplan–Meier survival curves showed the survival differences between high and low expression (grouping by median expression value) of the eight prognostic lncRNAs (A) and eight prognostic miRNAs (B).

*AFAP1-AS1*),which may be associated with survival time. The medium expression values of the lncRNAs were calculated and used as the threshold to divide the patients into two groups, one with upregulated and the other with downregulated genes. The subsequent KM survival curves demonstrated that all eight lncRNAs, especially *ST7-AS2* (ST7 Antisense RNA 2), were significantly associated with the overall survival time in EC patients (Fig 3A). Similarly, eight differentially expressed miRNAs and their KM survival analyses revealed a shorter overall

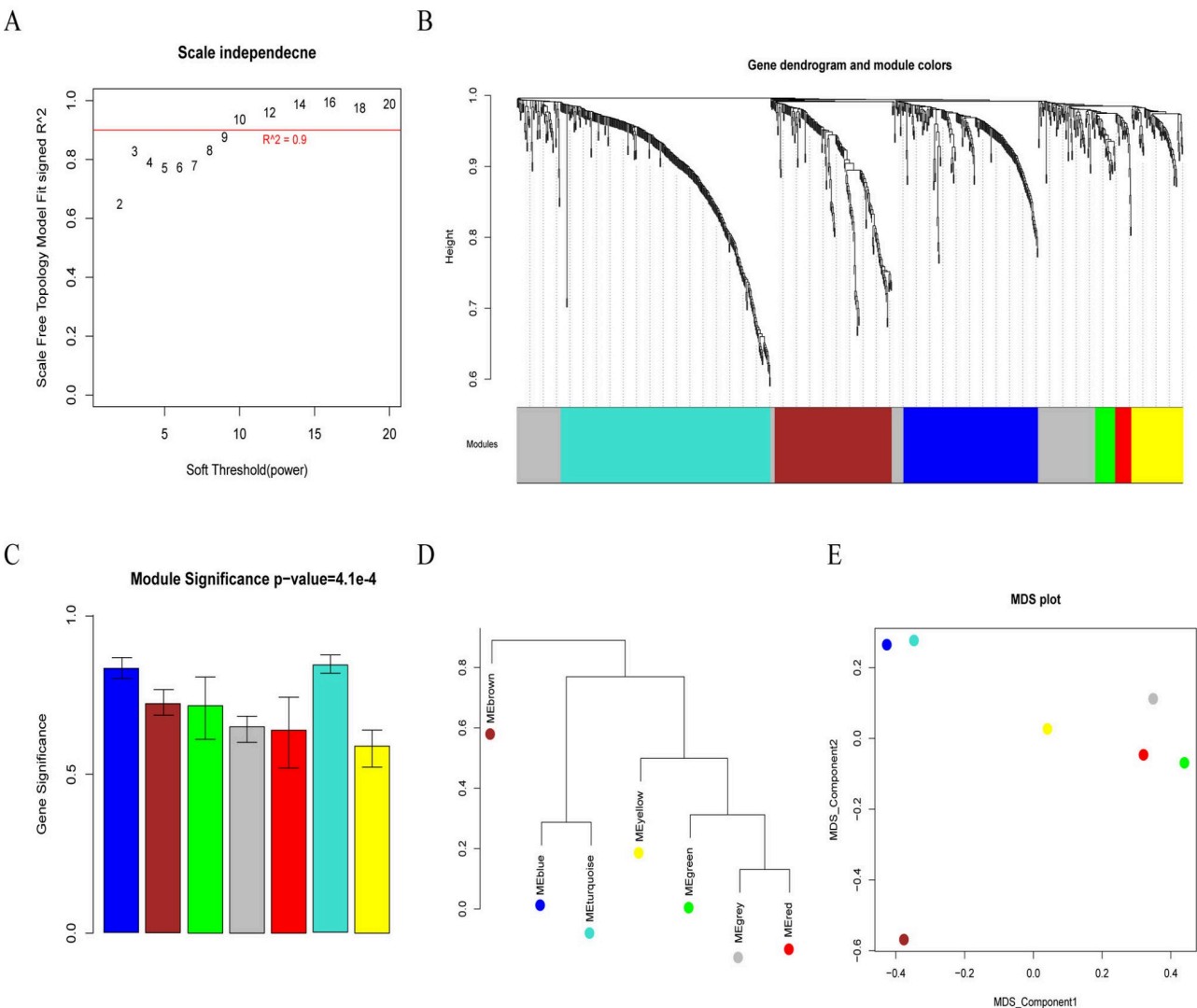

**Fig 4. Identification of the gene modules by weighted gene correlation network analysis (WGCNA).** A. Determination of parameter $\beta$ of the adjacency function in the WGCNA algorithm. The higher the square value of the correlation coefficient is, the closer the network is to the distribution without network scale. B. Hierarchical cluster analysis dendrogram of gene modules along with corresponding color assignments. Each color represents a certain gene module C. The correlation between trait values of each module and disease phenotype. Y-axis represent gene significance value, and the gene modules with higher gene significance value were significantly associated with disease phenotype. D. The hierarchical cluster analysis of different modules showed the correlations among different modules. E. The principal component analysis (PCA) analysis of different modules.

survival time in EC patients with higher expression of hsa-miR-1293, hsa-miR-135b, hsa-miR-299, hsa-miR-340, hsa-miR-412, hsa-miR-421, hsa-miR-1269, and hsa-miR-627 (Fig 3B).

## Construction of weighted co-expression network and identification of key modules associated with EC progression

In the present study, WGCNA package in R was used to group the DEGs with analogous expression profiles into modules by the Dynamic Hybrid cut hierarchical clustering. First, the power of β = 10 (scale-free R2 = 0.9) was selected as the soft-thresholding, for satisfying the scale-free criterion (Fig 4A). Subsequently, the gene modules were identified based on the

**Table 2. The correlation between the traits values of each module and the disease phenotype.**

| Color | Gene Count | Correlation |
|---|---|---|
| blue | 183 | 0.8 |
| brown | 159 | 0.68 |
| green | 27 | 0.67 |
| grey | 160 | 0.62 |
| red | 22 | 0.6 |
| turquoise | 285 | 0.82 |
| yellow | 70 | 0.57 |

dissimilarity coefficient and the following criteria: at least 30 genes in each gene module and cut height = 0.9. The different modules were clustered based on their trait values and the close ones were merged to a new module. Finally, a total of seven fused modules different from each other were obtained, which are represented by a unique color (Fig 4B). The correlation between the trait values of each module and the disease phenotype (alcohol-related and non-alcohol-related) were evaluated. As shown in Fig 4C and Table 2, the correlation coefficient ranged from 0.6 to 0.8 and the correlation $p < 0.05$. Of all the generated modules, the blue module was the most disease- related module, followed by the turquoise module. The hierarchical cluster analysis of different modules, principal component analysis (PCA) analysis based on the specific eigengene value, and significance of correlation with disease phenotype showed that the blue and turquoise modules were closely related with each other (Fig 4D and 4E). As a result, more attention was paid to the genes in the blue and turquoise modules in the subsequent studies.

## Prediction of lncRNA–miRNA interactions associated with EC development

To identify the presence of interplay between lncRNAs and miRNAs, the lncRNA–miRNA interactions were retrieved from the miRcode and starBase databases. All the screened DELs were uploaded to both miRcode and starBase databases to predict lncRNA–miRNA interactions. A total of 397 and 429 lncRNA–miRNA interactions were collected from the miRcode and starBase databases, respectively. Totally, 487 lncRNA–miRNA interactions were extracted from these two database after removal of redundancy. Then, 44 lncRNA–miRNA interactions involving DEMs were screened from the 487 lncRNA–miRNA interactions. The 44 lncRNA–miRNA interactions consisted of 13 DELs and 12 DEMs (Fig 5).

## Prediction of miRNA–target gene interactions associated with EC

*In silico* identification of lncRNA–miRNA interactions relevant to EC progression revealed 12 DEMs. To further predict miRNA–target gene interactions associated with EC, all the potential targets of the 12 DEMs were retrieved from the miRTarBase database. Subsequently, the miRNA-target gene interactions involving DEGs in blue and turquoise modules were screened to construct miRNA-target gene networks, respectively (Fig 6). The same nine miRNAs (*hsa-miR-1*, *hsa-miR-133a*, *hsa-miR-143*, h*sa-miR-16*, *hsa-miR-205*, *hsa-miR-33a*, *hsa-miR-503*, *hsa-miR-7*, and *hsa-miR-98*) were predicted to target DEGs in in blue and turquoise modules respectively. Fig 6A showed the miRNA-target gene network consisting of nine miRNAs and their targeted 132 DEGs in blue module. While Fig 6B showed the miRNA-target gene network consisting of nine miRNAs and their targeted 198 DEGs in turquoise module.

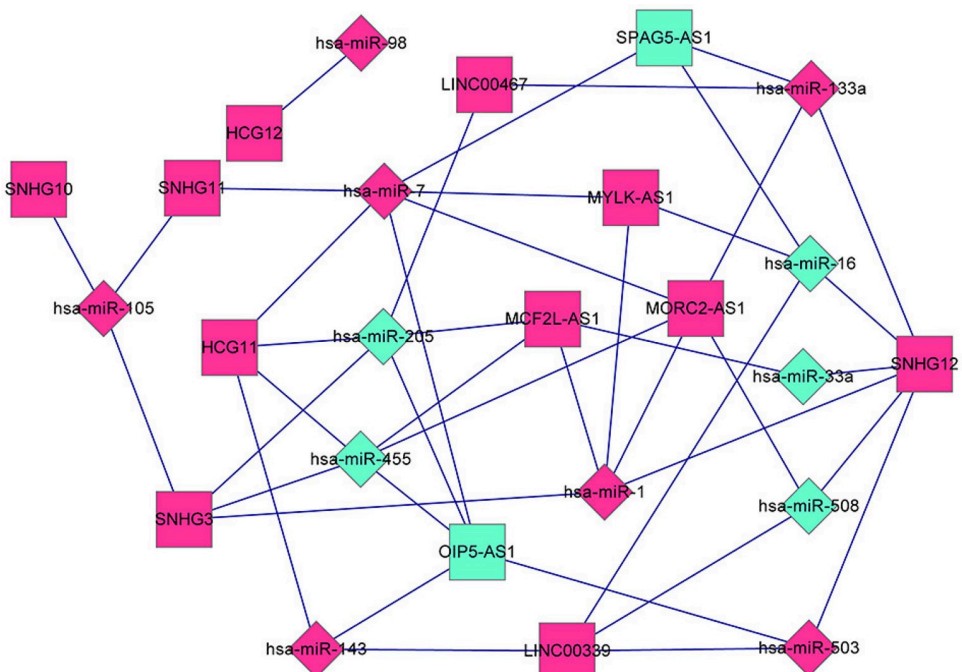

**Fig 5. DEL–DEM regulatory network in alcohol-related EC progression.** All the screened DELs were uploaded to both miRcode and starBase databases to predict lncRNA–miRNA interactions, of which 44 lncRNA–miRNA interactions involving DEMs were screened. The network was visualized based on the 44 lncRNA–miRNA interactions. Diamond and square nodes represent miRNAs and lncRNAs, respectively. Red and green colors represent upregulation and downregulation, respectively. DEL, differentially expressed lncRNA; DEM, differentially expressed miRNA.

## Construction of ceRNA networks and enrichment analysis of the target DEGs associated with EC progression

The ceRNA networks associated with EC development were constructed by integrating the lncRNA–miRNA and miRNA–mRNA interactions, which led to establishment of two ceRNA regulator networks. The results of our analyses revealed that the ceRNA network generated with the blue module (Fig 7A) consisted of 12 DELs (*SPAG5-AS1*, *OIP5-AS1*, *SNHG3*, *SNHG11*, *HCG11*, *MORC2-AS1*, *SNHG12*, LINC00467, *MCF2L-AS1*, *MYLK-AS1*, LINC00339, and *HCG12*); 9 DEMs (hsa-miR-1, hsa-miR-133a, hsa-miR-143, hsa-miR-16, hsa-miR-205, hsa-miR-33a, hsa-miR-503, hsa-miR-7, and hsa-miR-98); 132 DEGs (64 downregulated and 68 upregulated ones); and the nodes with top 15 degrees (such as hsa-miR-16, hsa-miR-7, hsa-miR-1, hsa-miR-205, hsa-miR-33a, hsa-miR-133a, hsa-miR-143, hsa-miR-98, hsa-miR-503, *GALNT1*, *GLS*, *SNHG12*, *FERMT2*, *ATP1B*, and *OGT*) (Table 3). A total of 198 DEGs (98 downregulated and 100 upregulated ones) in the turquoise module as well as the above 12 DELs and 9 DEMs were involved in the second ceRNA network (Fig 7B), and the nodes with top 15 degrees included hsa-miR-16, hsa-miR-1, hsa-miR-7, hsa-miR-143, hsa-miR-205, hsa-miR-98, hsa-miR-33a, hsa-miR-133a, hsa-miR-503, *ALCAM*, *SNHG12*, *SMAP1*, *TIMM23*, *GPR107*, and *ABCC5* (Table 3).

Moreover, enrichment analysis revealed that the target DEGs in the blue and turquoise modules were highly associated with four KEGG pathways such as other types of O-glycan bio-synthesis, N-glycan biosynthesis, and proteoglycans in cancer and 35 functional categories including zinc ion binding, extracellular exosome, leukocyte migration, cell surface, and mitotic nuclear division (Fig 8 and Table 4). Remarkably, the gene *ST6GAL1* (β-galactoside

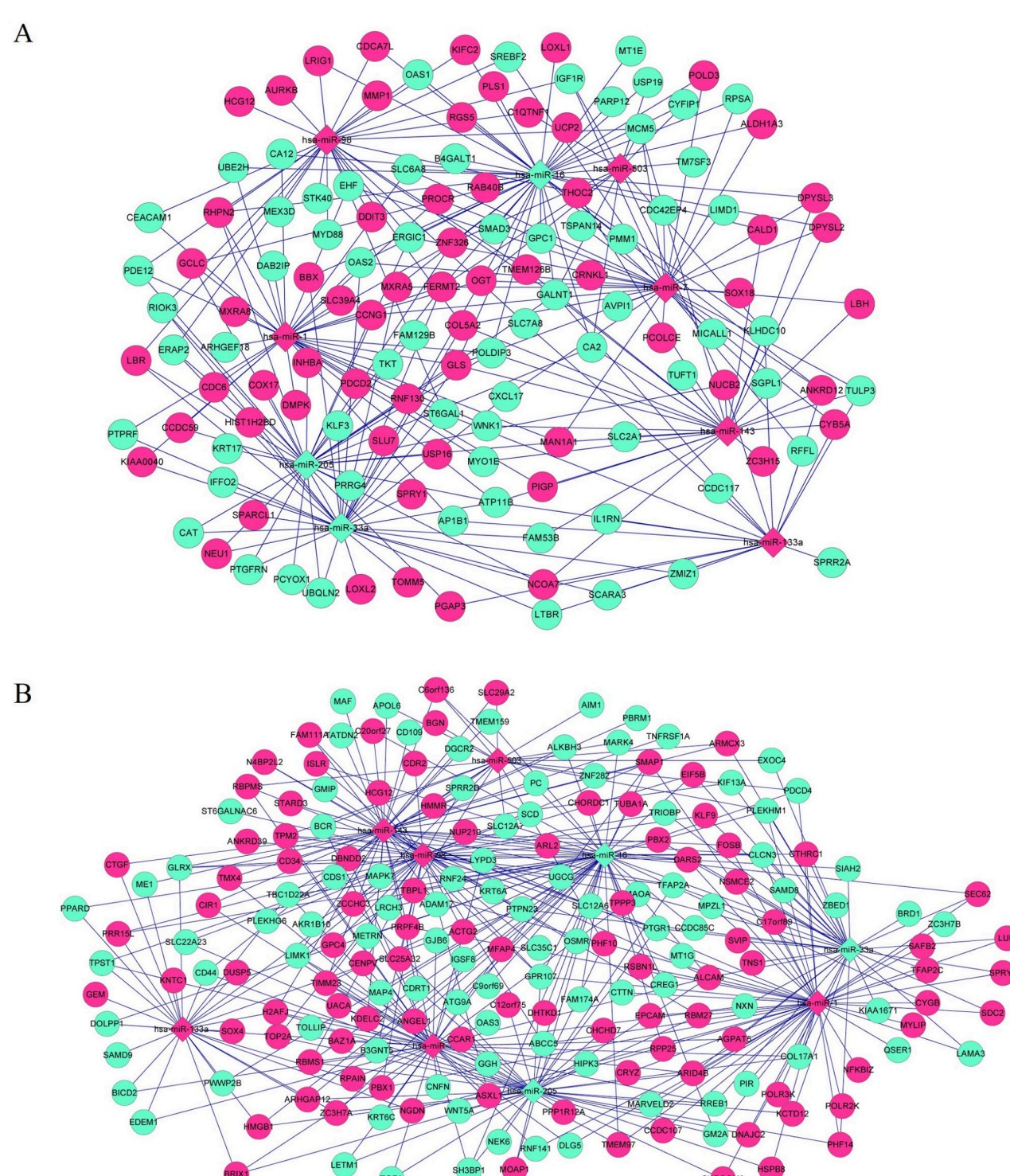

**Fig 6. DEM–DEG regulatory network in alcohol-related EC progression.** The 12 DEMs in DEL–DEM regulatory network were uploaded to miRTarBase database to predict their targeted genes, and the miRNA-target gene interactions involving DEGs in blue (A) and turquoise (B) modules were screened to construct miRNA-target gene networks, respectively. Diamond and round nodes represent miRNAs and target genes, respectively. Red and green colors represent upregulation and downregulation, respectively. DEM, differentially expressed miRNAs; DEG, differentially expressed genes.

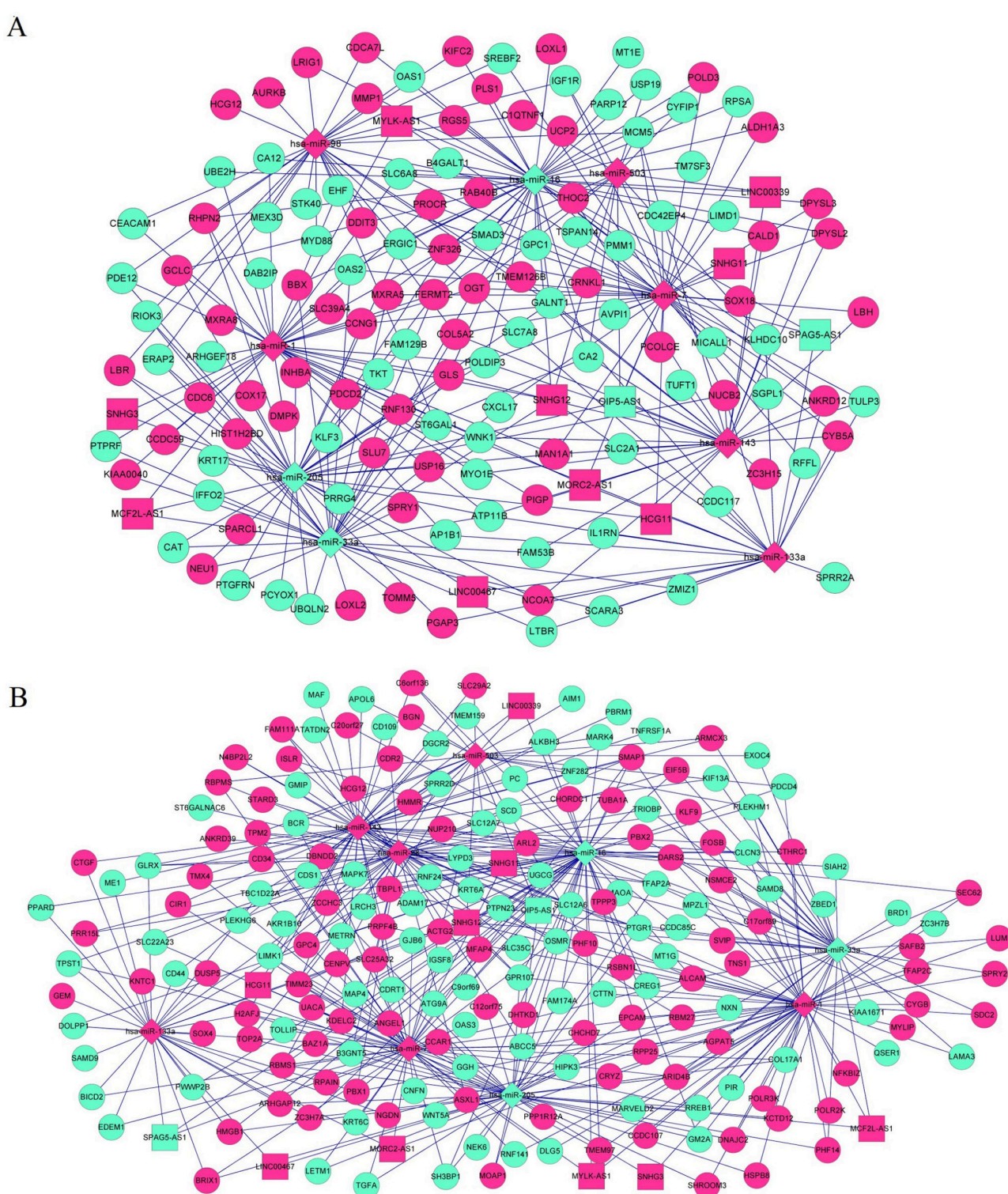

**Fig 7. CeRNA regulatory network in alcohol-related EC progression.** The lncRNA–miRNA interactions, and miRNA-target gene interactions were integrated as lncRNA-miRNA-target genes interactions, and ceRNA network was visualized based on lncRNA-miRNA-target genes interactions. A. The ceRNA network constructed by the DEGs in the blue module. B. The ceRNA network constructed by the DEGs in the turquoise module. Diamond, square, and round nodes represent miRNAs, lncRNAs, and mRNA, respectively. Red and green colors represent upregulation and downregulation, respectively.

**Table 3. The top 15 nodes in the competing endogenous RNA (ceRNA) regulatory network in the blue module and the turquoise module.**

| Module | Nodes | Description | Degree |
|---|---|---|---|
| Blue module | hsa-miR-16 | miRNA | 47 |
| | hsa-miR-7 | miRNA | 39 |
| | hsa-miR-1 | miRNA | 39 |
| | hsa-miR-205 | miRNA | 36 |
| | hsa-miR-33a | miRNA | 35 |
| | hsa-miR-143 | miRNA | 30 |
| | hsa-miR-98 | miRNA | 29 |
| | hsa-miR-133a | miRNA | 24 |
| | hsa-miR-503 | miRNA | 15 |
| | GALNT1 | mRNA | 6 |
| | GLS | mRNA | 6 |
| | SNHG12 | lncRNA | 5 |
| | FERMT2 | mRNA | 5 |
| | ATP11B | mRNA | 5 |
| | OGT | mRNA | 5 |
| Turquoise module | hsa-miR-16 | miRNA | 57 |
| | hsa-miR-1 | miRNA | 55 |
| | hsa-miR-7 | miRNA | 54 |
| | hsa-miR-143 | miRNA | 49 |
| | hsa-miR-205 | miRNA | 47 |
| | hsa-miR-98 | miRNA | 46 |
| | hsa-miR-33a | miRNA | 39 |
| | hsa-miR-133a | miRNA | 31 |
| | hsa-miR-503 | miRNA | 24 |
| | ALCAM | mRNA | 6 |
| | SNHG12 | lncRNA | 5 |
| | SMAP1 | mRNA | 5 |
| | TIMM23 | mRNA | 5 |
| | GPR107 | mRNA | 5 |
| | ABCC5 | mRNA | 5 |

hsa, Homo sapiens; miR, microRNA; lncRNAs, long non coding RNAs.

α2,6 sialyltranferase 1), regulated by hsa-miR-1, hsa-miR-133a, hsa-miR-33a, and hsa-miR-98, was involved in both other types of O-glycan biosynthesis and N-glycan biosynthesis pathways, and should therefore be further investigated in detail in further studies.

## Discussion

EC is one of the leading causes of cancer-related deaths with patients exhibiting relatively poor prognosis. Alcohol consumption is a major risk factor for EC development. However, high throughput screening of the genetic alterations between patients with alcohol-related and non-alcohol-related EC has been rarely performed. The current study analyzed the RNA-seq and miRNA-seq data downloaded from TCGA database to identify the aberrantly expressed genes, miRNAs, and lncRNAs in patients with alcohol-related EC. In addition, multiple-level interactions among these molecules were analyzed and ceRNA networks constructed to identify the alcohol-related EC-associated transcriptional RNA interactions and thus reveal the potential post-transcriptional regulatory mechanisms of significance in alcohol-related EC.

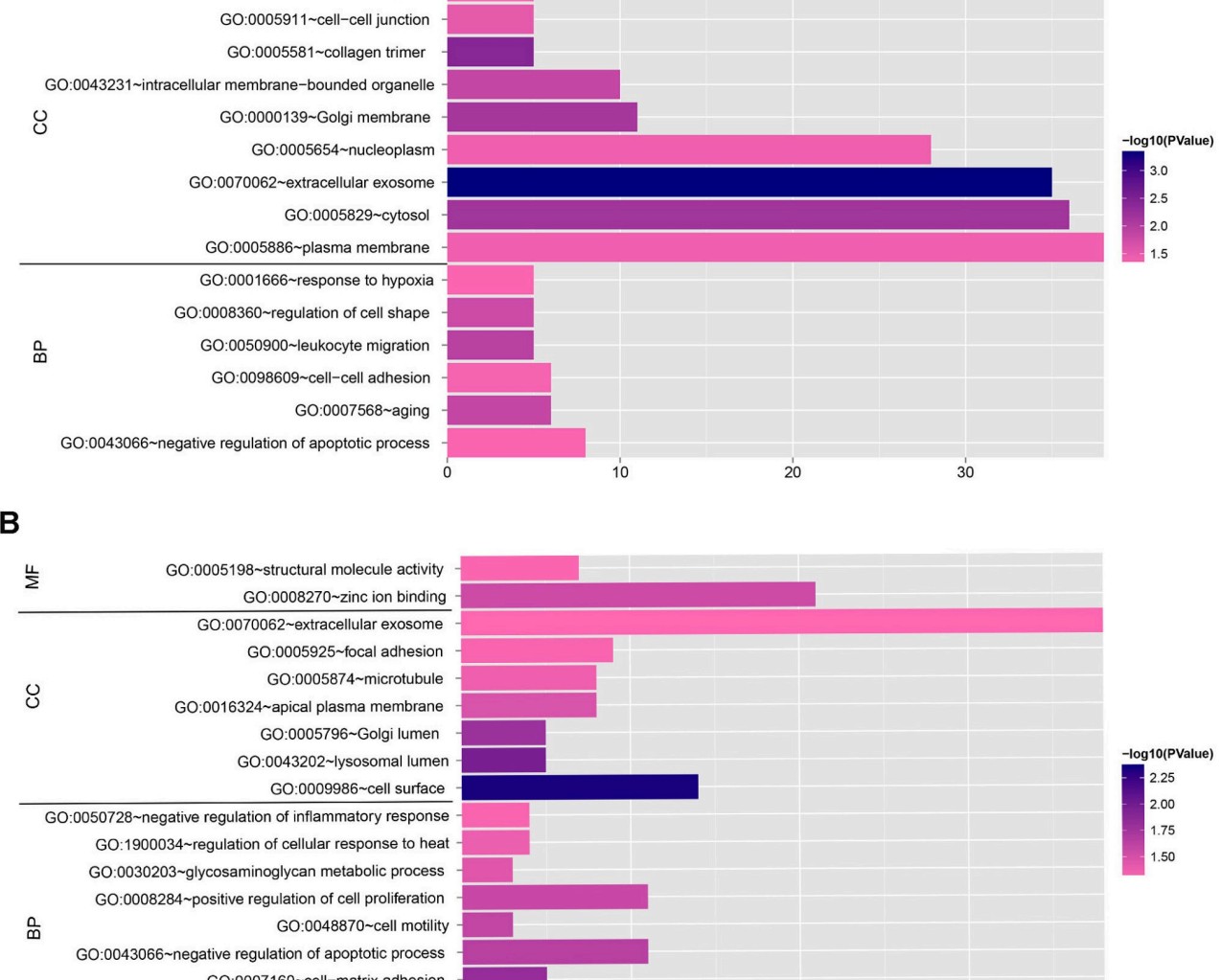

**Fig 8. Gene Ontology enrichment analysis of the genes in the ceRNA networks associated with alcohol-related EC.** A. The analysis of genes in the blue module. B. The analysis of genes in the turquoise module. The Gene Ontology annotation terms consist of molecular function (MF) annotation terms, cellular component (CC) annotation terms and biological process (BP) annotation terms. The length of column represent gene numbers enriched in this annotation term.

By differential expression analysis, a total of 906 DEGs, 40 DELs, and 52 DEMs were identified between the patients with alcohol-related and non-alcohol-related EC. In addition, univariate cox regression identified eight prognostic lncRNAs (*C17orf100*, *RNU11*, *MORC2-AS1*, *SNHG10*, *ASMTL-AS1*, *ST7-AS2*, *MIR210HG*, and *AFAP1-AS1*) and eight miRNAs (hsa-miR-1269, hsa-miR-421, hsa-miR-340, hsa-miR-1293, hsa-miR-135b, hsa-miR-299, hsa-miR-412, and hsa-miR-627). Among these, hsa-miR-1269 and ST7-AS2 may be associated with alcohol-related EC. It has been reported that miR-1269 is overexpressed in many types of cancers, such

**Table 4. The KEGG pathway enrichment analysis of the genes in the ceRNA networks.**

| Term | Count | P Value | Genes |
|---|---|---|---|
| Genes in the blue module | | | |
| hsa00514: Other types of O-glycan biosynthesis | 3 | 0.020876 | B4GALT1, ST6GAL1, OGT |
| hsa00510: N-Glycan biosynthesis | 3 | 0.048637 | B4GALT1, ST6GAL1, MAN1A1 |
| hsa04520: Adherens junction | 3 | 0.093067 | IGF1R, PTPRF, SMAD3 |
| Genes in the turquoise module | | | |
| hsa05205: Proteoglycans in cancer | 7 | 0.010908 | WNT5A, CTTN, CD44, LUM, PPP1R12A, PDCD4, SDC2 |

hsa, Homo sapiens.

as hepatocellular carcinoma and lung cancer and can further regulate cell proliferation, apoptosis, epithelial-mesenchymal transition, and metastasis [38, 39]. It is hypothesized that miR-1269 could serve as a potential prognostic biomarker and therapeutic target in these cancers. However, the expression of miR-1269 and its role in EC has not been elucidated yet. *ST7-AS2* (ST7 antisense RNA 2) is an lncRNA, and various studies have been performed to investigate the association between *ST7-AS2* and cancer [40, 41]. Consequently, further studies should focus on the role of miR-1269 and *ST7-AS2* in the development of alcohol-related EC.

WGCNA is a powerful 'guilt-by-association'-based method to extract gene modules from large heterogeneous RNA expression datasets. In the present study, we identified seven fused modules and DEGs in two relevant modules (the blue and turquoise modules), which were then selected for constructing the ceRNA networks. Finally, two ceRNA networks of transcriptional RNA interactions, which may play critical roles in alcohol-related EC, were built comprising lncRNAs such as *SPAG5-AS1*, *OIP5-AS1*, *SNHG3*, *SNHG11*, *HCG11*, *MORC2-AS1*, *SNHG12*, LINC00467, *MCF2L-AS1*, *MYLK-AS1*, LINC00339, and *HCG12*. Sperm-associated antigen 5 (SPAG5) has been recently identified as a prognostic biomarker for breast and cervical cancers [42, 43]. The lncRNA *OIP5-AS1* has been shown to modulate Bcl-2 expression by targeting miR-448 in lung adenocarcinoma cells and is inversely related to the patient survival rate [44]. The lncRNAs such as *SNHG3*, *SNHG11*, and *SNHG12* belong to the family of small nucleolar RNA host genes *(SNHGs)*. In hepatocellular carcinoma patients, associations between increased SNHG3 levels with malignant status and poor prognosis have been reported [44]. Moreover, *SNHG12* could promote the proliferation of gastric carcinoma cells BGC-823 by targeting miRNA-199a/b-5p [40]. Interestingly, *SNHG12* was also identified to be a hub in both blue and turquoise modules in our study. As a result, it is only rational that these cancer-related lncRNAs be investigated in further studies for their role in EC development. Among the 12 lncRNAs identified in our study, *MORC2-AS1* was significantly associated with the overall survival rate of patients with EC. Remarkably, the ceRNA network analysis revealed the significance of the MORC2-AS1–hsa-miR-7–KRT6C/GJB6 interaction in EC progression. *KRT6C* and *GJB6* are two genes that we were among the top 10 selected DEGs. miR-7 has been reported to be overexpressed in ESCC and associated with its differentiation [45]. Furthermore, miR-7 upregulation has been documented in the brain of human alcoholics [46]. Consequently, the MORC2-AS1–hsa-miR-7–KRT6C/GJB6 interaction may play an important role in the development of alcohol-related EC. Functional enrichment analysis of the genes in the ceRNA networks indicated that the pathways O-glycan biosynthesis, N-glycan biosynthesis, and proteoglycans in cancer, may be involved in the development of alcohol-related EC. Alcohol consumption has been reported to result in Golgi fragmentation, which is the central station of glycosylation [47–49], and decreased glycosyltransferase as well as kinase activities [49, 50]. Furthermore, chronic alcohol administration has contributed to glycoprotein

accumulation [51]. Meanwhile, previous studies have demonstrated that Golgi apparatus fragmentation is associated with pro-oncogenic and pro-metastatic pathways [52]. Moreover, ethanol has been shown to impair N-linked glycosylation by influencing dolichol biosynthesis and lead to impaired dolichol-linked oligosaccharide assembly [53]. N-glycosylation induces various functional changes of glycoproteins, including cell surface receptors as well as adhesion molecules, and regulates tumor cell proliferation and metastasis [54, 55]. N-glycosylation inhibition by N-acetylglucosamine transferase contributes to ESCC treatment [56, 57]. N-glycans obtained from haptoglobin can serve as effective EC biomarkers [58–60]. Therefore, alcohol may be involved in EC progression by affecting N-glycan biosynthesis. ST6GAL1 (ST6 beta-galactoside alpha-2, 6-sialyltransferase 1), enriched in the N-glycan biosynthesis pathway, is an enzyme catalyzing the h-2, 6 sialylation on N-glycans. ST6GAL1 mediates the addition of 2,6-linked sialic acid to glycoproteins in the Golgi apparatus [61]. Alcohol regulated *ST6GAL1* gene downregulation leads to defective apolipoprotein glycosylation, followed by alcoholic steatosis [62]. Moreover, *ST6GAL1* is associated with alcohol-mediated liver injury [63]. Further, the altered expression of *ST6GAL1* has been detected in many types of cancers [64]. A genetic variant of *ST6GAL1* has also been associated with EC progression [65–67]. *ST6GAL1*, as a risk gene, has been shown to be involved in protein glycosylation associated with the molecular mechanisms of EC [68]. Moreover, its expression could be regulated by hsa-miR-1 (targeted by the lncRNAs *MCF2L-AS1*, *MORC2-AS1*, *MYLK-AS1*, *SNHG12*, and *SNHG3*), hsa-miR-133a (targeted by the lncRNAs LINC00467, *MORC2-AS1*, *SNHG12*, and *SPAG5-AS1*), hsa-miR-33a (targeted by the lncRNAs *MCF2L-AS1* and *SNHG12*), and hsa-miR-98 (targeted by the lncRNA *HCG12*) as per the outcomes of our interaction analysis. As discussed previously, the lncRNAs *SNHG12*, *SNHG3*, and *SPAG5*-AS1 have been reported to be associated with the development of many types of cancer. Furthermore, miR-1 could suppress the proliferation and promote the apoptosis of EC cells [69]. miR-133a suppresses esophageal cancer cell migration and invasion and can serve as an effective prognostic biomarker in EC [70, 71]. Additionally, miR-33a was found to be upregulated in the co-cultured media of fibroblasts and EC cells [72]. Furthermore, *hsa-miR-1*, *hsa-miR-133a*, *hsa-miR-33a*, and *SNHG12* were hub genes in the ceRNA networks generated in our study, indicating their possible significant association with EC progression. In this context, the SNHG12–miR-1–ST6GAL1, SNHG3–miR-1–ST6GAL1, SPAG5-AS1–miR-133a–ST6GAL1, and SNHG12–hsa-miR-33a–ST6GAL1 interactions may play important roles in alcohol-related EC.

In this paper, the molecular expression characteristics of alcoholic and non-alcoholic EC were systematically analyzed, and some lncRNAs, miRNAs and mRNAs that significantly correlated with clinical characteristics and overall survival of patients were screened out. In addition, the regulatory relationship among these key molecules was investigated. This study provides targets to investigate the pathological mechanism of alcoholic-related EC at molecular level. However, these investigations were concluded based on bioinformatics analysis and warrant further validation for clinical application via comprehensive *in vitro* and *in vivo* studies involving immunohistochemistry or quantitative real-time polymerase chain reaction.

Studies have demonstrated that identification of disease-associated miRNAs/lncRNAs has been will contribute to understand the molecular pathogenesis and to develop molecular tools for the prevention, diagnosis and treatment of diseases [73, 74]. In this study, we identified several miRNAs and lnRNAs that significantly associated with alcohol-related EC by using correlation analysis with clinical characteristics and prognosis. In fact, many computational models were developed to identify disease-related miRNAs and lncRNAs [27–29]. For example, Chen et al. developed a computational model of Matrix Decomposition and Heterogeneous Graph Inference for miRNA-disease association prediction [29] and developed the method of Laplacian Regularized Least Squares for LncRNA–Disease Association (LRLSLDA) in the

semisupervised learning framework [28], these computational models have been confirmed to be effective. Additionally, computational models have been considered as important biological tool for biomedical research, which help to identify the most important miRNAs/lncRNAs-disease associations for experimental validation, markedly decreasing the cost and time for biological experiments [73, 74]. Therefore, these computational models should be used for validation our findings and for identifying other potential disease-associated miRNAs and lncRNAs in the future. Similarly, we investigated lncRNA-miRNA interactions based on miR-code and starBase databases. While various network algorithms or models have been developed to predict lncRNA-miRNA interactions, which showed superior prediction performance than others [75–77]. In order to screen the most important lncRNA-miRNA interactions, different algorithms or methods should be used in the future. Moreover, circRNAs also play crucial roles in the occurrence and progression of many complex diseases, and several state-of-the-art computational models for predicting circRNA-disease associations have also been developed [78]. Thus, investigation of circRNAs that associated with alcohol-related EC using these computational models should be the future direction of our work.

## Conclusion

The results of the present study have revealed significant lncRNA–miRNA–mRNA regulatory interactions possibly involved in the progression of alcohol-related EC. These ceRNA networks thus provide a novel insight into the molecular mechanism underlying alcohol-related EC development. However, as these results were based only on *in silico* predictions, further *in vivo* and *in vitro* experiments are needed to validate these findings.

## Supporting information

**S1 Table. Association of miRNAs with clinical features in EC.**
(DOCX)

**S2 Table. Association of mRNAs with clinical features in EC.**
(DOCX)

**S3 Table. Association of lncRNAs with clinical features in EC.**
(DOCX)

## Author Contributions

**Conceptualization:** Yu Wang.

**Writing – original draft:** Quan Du, Ren-Dong Xiao, Rong-Gang Luo, Jin-Bao Xie, Zu-Dong Su, Yu Wang.

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
