## [Decision Letter · Decision Letter 0]

28 Sep 2021

PONE-D-21-25185Construction of long non-coding RNA-and microRNA-mediated competing endogenous RNA networks in alcohol-related esophageal cancerPLOS ONE

Dear Mr. Yu Wang,

Thank you for submitting your manuscript to PLOS ONE. After careful consideration, we feel that it has merit but does not fully meet PLOS ONE’s publication criteria as it currently stands. Therefore, we invite you to submit a revised version of the manuscript that addresses the points raised during the review process.

We look forward to receiving your revised manuscript.

Kind regards,

Qi Zhao

Academic Editor

PLOS ONE

Journal Requirements:

"There are no financial conflicts of interest to disclose."

5. Please include your tables as part of your main manuscript and remove the individual files. Please note that supplementary tables (should remain/ be uploaded) as separate "supporting information" files

Reviewers' comments:

Reviewer's Responses to Questions

**Comments to the Author**

1. Is the manuscript technically sound, and do the data support the conclusions?

Reviewer #1: Partly

Reviewer #2: Yes

2. Has the statistical analysis been performed appropriately and rigorously? 

Reviewer #1: No

Reviewer #2: Yes

3. Have the authors made all data underlying the findings in their manuscript fully available?

Reviewer #1: Yes

Reviewer #2: Yes

4. Is the manuscript presented in an intelligible fashion and written in standard English?

Reviewer #1: No

Reviewer #2: No

5. Review Comments to the Author

Reviewer #1: In this paper, authors downloaded the RNA profiles of patients with alcohol-related and non-alcohol-related EC from The Cancer Genome Atlas (TCGA) database to systematically investigate the lncRNA–miRNA–mRNA interactions related to EC using bioinformatics methods. Identifying susceptible biomarkers can help in better understanding the etiology of EC and then improve the management of this disease. The following are the detailed comments:

1. Spelling and grammar mistakes should be carefully checked in the manuscript.

2. Many computational models were developed to identify disease-related miRNAs and lncRNAs. Authors should introduce some important computational models for them as the future direction of this study (PMIDs: 29939227, 24002109, and 30142158).

3. The method used to predict lncRNA-miRNA interactions and miRNA–target gene interactions should be introduced in more detail in the “Result” section.

4. Each formula presented in the manuscript should be given a corresponding number.

5. Each figure presented in the manuscript is not described in detail.

6. Important reviews about disease-miRNA association prediction and disease-lncRNA association prediction should be emphatically introduced (PMIDs: 29045685 and 27345524).

7. The novelty of this work is not well highlighted. In addition, limitations of the work and work prospects were not provided.

8. I want to know whether the framework proposed in this manuscript is effective for other cancers.

Reviewer #2: In the current manuscript, the authors explored the lncRNA–miRNA–mRNA networks involved in alcohol-related esophageal cancer (EC). They downloaded RNA sequencing and clinical data from TCGA. The lncRNA–miRNA–mRNA competing endogenous RNA (ceRNA) networks were constructed based on the information from miRcode, starBase, and miRTarBase databases. A total of 906 DEGs, 40 DELs, and 52 DEMs were identified. There were eight lncRNAs and eight miRNAs, especially ST7-AS2 and miR-1269, which were significantly associated with the survival rate of patients with EC. Besides, SNHG12–miR-1–ST6GAL1, SNHG3–miR-1–ST6GAL1, SPAG5-AS1–miR-133a–ST6GAL1, and SNHG12–hsa-miR-33a–ST6GA interactions, related to the N-Glycan biosynthesis pathway, may have key roles in alcohol-related EC. However, there are some problems to be further improved before acceptance for publication.

1. Literature review is somewhat incomplete in the introduction, especially about the progress of the bioinformatic analysis about the discovery of ncRNA related with EC.

2. The flowchart in Fig.1 is too simple to show the process very well, please improve it. Also the authors should put Fig.1 before Fig.2.

3. In the Data acquisition section, the authors stated that they downloaded the sequencing data from the TCGA database. Is the authors sure about this? Usually we just download the Gene Expression Quantification data. If the authors processed the raw sequencing data, please write clearly the processing process.

4. Also about the Data acquisition. What is the “paired samples” mean? Were 18 normal samples used in the study? Please make these clearer.

5. Why is the criteria of DEG set to false discovery rate (FDR) < 0.05 and |fold change (FC)| > 1.5?

6. There are some misspellings of software names, such as “miRcorde”, “miRecorde”, “Cystoscope”. Please check these issues carefully.

7. I suggest the authors should discuss more about the meaning of their research in the last section, as they use several online tools.

8. Could the authors give some discussions whether their method and obtained result could be further used to predict EC-related ncRNAs? It could be considered as the future direction of their work. Some recommended studies are helpful (PMIDs: 33588070, 34232474, 34329377; DOI: 10.1016/j.knosys.2019.105261).

9. English expressions need to be edited more careful and more native, in this manuscript, there are some mistakes.

6. PLOS authors have the option to publish the peer review history of their article (what does this mean?). If published, this will include your full peer review and any attached files.

Reviewer #1: **Yes: **Nana Guan

Reviewer #2: No

---

## [Author Response · Author response to Decision Letter 0]

18 Nov 2021

Dear editor,

We would like to extend our most sincere gratitude to the reviewers' comments on our manuscript entitled "Construction of long non-coding RNA-and microRNA-mediated competing endogenous RNA networks in alcohol-related esophageal cancer ". Those comments are all valuable and very helpful for revising and improving our paper, as well as the essential guiding significance to our researches. We have revised the comments extensively and have made the corrections in order to meet with approval. The leading corrections in the paper and the response to the reviewers' comments are as follows.

We appreciate for Editors/Reviewers' warm work earnestly and hope that the correction will meet with approval.

Once again, thanks for the comments and suggestions.

Sincerely,

Quan Du

A point-by-point response to reviewer’s comments

Reviewer #1

In this paper, authors downloaded the RNA profiles of patients with alcohol-related and non-alcohol-related EC from The Cancer Genome Atlas (TCGA) database to systematically investigate the lncRNA–miRNA–mRNA interactions related to EC using bioinformatics methods. Identifying susceptible biomarkers can help in better understanding the etiology of EC and then improve the management of this disease. The following are the detailed comments:

Comment 1: Spelling and grammar mistakes should be carefully checked in the manuscript.

Response: Thanks for your valuable comment. The mistakes have been checked and corrected in this revision. In addition, we have sent this manuscript to be edited by professional editors at Editage. The language editing certificate has been provided.

Comment 2: Many computational models were developed to identify disease-related miRNAs and lncRNAs. Authors should introduce some important computational models for them as the future direction of this study (PMIDs: 29939227, 24002109, and 30142158)

Response: Thanks for your valuable comment. We have discussed these computational models in the last paragraph, as follows:

Studies have demonstrated that identification of disease-associted miRNAs/lncRNAs has been will contribute to understand the molecular pathogenesis and to develop molecular tools for the prevention, diagnosis and treatment of diseases (73, 74). In this study, we identified several miRNAs and lnRNAs that significantly associated with alcohol-related EC by using correlation analysis with clinical characteristics and prognosis. In fact, many computational models were developed to identify disease-related miRNAs and lncRNAs (27-29). For example, Chen et al. developed a computational model of Matrix Decomposition and Heterogeneous Graph Inference for miRNA-disease association prediction (29) and developed the method of Laplacian Regularized Least Squares for LncRNA–Disease Association (LRLSLDA) in the semisupervised learning framework (28), these computational models have been confirmed to be effective. Additionally, computational models have been considered as important biological tool for biomedical research, which help to identify the most important miRNAs/lncRNAs-disease associations for experimental validation, markedly decreasing the cost and time for biological experiments (73, 74). Therefore, these computational models should be used for validation our findings and for identifying other potential disease-associated miRNAs and lncRNAs in the future.

Comment 3: The method used to predict lncRNA-miRNA interactions and miRNA–target gene interactions should be introduced in more detail in the “Result” section.

Response: Thanks for your valuable comment. We have revised the descriptions in Result section, as follows:

Prediction of lncRNA–miRNA interactions associated with EC development

To identify the presence of interplay between lncRNAs and miRNAs, the lncRNA–miRNA interactions were retrieved from the miRcode and starBase databases. All the screened DELs were uploaded to both miRcode and starBase databases to predict lncRNA–miRNA interactions. A total of 397 and 429 lncRNA–miRNA interactions were collected from the miRcode and starBase databases, respectively. Totally, 487 lncRNA–miRNA interactions were extracted from these two database after removal of redundancy. Then, 44 lncRNA–miRNA interactions involving DEMs were screened from the 487 lncRNA–miRNA interactions. The 44 lncRNA–miRNA interactions consisted of 13 DELs and 12 DEMs (Fig. 5).

Prediction of miRNA–target gene interactions associated with EC

In silico identification of lncRNA–miRNA interactions relevant to EC progression revealed 12 DEMs. To further predict miRNA–target gene interactions associated with EC, all the potential targets of the 12 DEMs were retrieved from the miRTarBase database. Subsequently, the miRNA-target gene interactions involving DEGs in blue and turquoise modules were screened to construct miRNA-target gene networks, respectively (Fig. 6). The same nine miRNAs (hsa-miR-1, hsa-miR-133a, hsa-miR-143, hsa-miR-16, hsa-miR-205, hsa-miR-33a, hsa-miR-503, hsa-miR-7, and hsa-miR-98) were predicted to target DEGs in in blue and turquoise modules respectively. Fig. 6A showed the miRNA-target gene network consisting of nine miRNAs and their targeted 132 DEGs in blue module. While Fig. 6B showed the miRNA-target gene network consisting of nine miRNAs and their targeted 198 DEGs in turquoise module.

Comment 4: Each formula presented in the manuscript should be given a corresponding number.

Response: Thanks for your valuable comment. We have added a corresponding number for the formula in the manuscript.

Comment 5: Each figure presented in the manuscript is not described in detail.

Response: Thanks for your valuable comment. The figure legends have been described in detail according to your suggestion, as follows:

Figure 2. Cluster analysis of the selected RNAs. 

A. The distribution of expression density of mRNAs, miRNAs and lncRNAs. The peak expression density increased after filtering the RNAs with expression level < 1. B. Clustering map of samples based on differential expression of genes, miRNAs, and lncRNAs. In which, the samples in one group tended to cluster together.

Figure 3. Prognosis related lncRNAs and miRNAs.

Univariate Cox regression for differentially expressed lncRNAs and miRNAs identified eight lncRNAs and eight miRNAs were associated with overall survival of patients with alcohol-related esophageal cancer. Kaplan–Meier survival curves showed the survival differences between high and low expression (grouping by median expression value) of the eight prognostic lncRNAs (A) and eight prognostic miRNAs (B).

Figure 4. Identification of the gene modules by weighted gene correlation network analysis (WGCNA). 

A. Determination of parameter β of the adjacency function in the WGCNA algorithm. The higher the square value of the correlation coefficient is, the closer the network is to the distribution without network scale. B. Hierarchical cluster analysis dendrogram of gene modules along with corresponding color assignments. Each color represents a certain gene module. C. The correlation between trait values of each module and disease phenotype. Y-axis represent gene significance value, and the gene modules with higher gene significance value were significantly associated with disease phenotype. D. The hierarchical cluster analysis of different modules showed the correlations among different modules. E. The principal component analysis (PCA) analysis of different modules.

Figure 5. DEL–DEM regulatory network. 

All the screened DELs were uploaded to both miRcode and starBase databases to predict lncRNA–miRNA interactions, of which 44 lncRNA–miRNA interactions involving DEMs were screened. The network was visualized based on the 44 lncRNA–miRNA interactions. Diamond and square nodes represent miRNAs and lncRNAs, respectively. Red and green colors represent upregulation and downregulation, respectively. DEL, differentially expressed lncRNA; DEM, differentially expressed miRNA.

Figure 6. DEM–DEG regulatory network. 

The 12 DEMs in DEL–DEM regulatory network were uploaded to miRTarBase database to predict their targeted genes, and the miRNA-target gene interactions involving DEGs in blue (A) and turquoise (B) modules were screened to construct miRNA-target gene networks, respectively. Diamond and round nodes represent miRNAs and target genes, respectively. Red and green colors represent upregulation and downregulation, respectively. DEM, differentially expressed miRNAs; DEG, differentially expressed genes.

Figure 7. CeRNA regulatory network.

The lncRNA–miRNA interactions, and miRNA-target gene interactions were integrated as lncRNA-miRNA-target genes interactions, and ceRNA network was visualized based on lncRNA-miRNA-target genes interactions. A. The ceRNA network constructed by the DEGs in the blue module. B. The ceRNA network constructed by the DEGs in the turquoise module. Diamond, square, and round nodes represent miRNAs, lncRNAs, and mRNA, respectively. Red and green colors represent upregulation and downregulation, respectively.

Figure 8. The Gene Ontology enrichment analysis of the genes in the ceRNA networks. 

A. The analysis of genes in the blue module. B. The analysis of genes in the turquoise module. The Gene Ontology annotation terms consist of molecular function (MF) annotation terms, cellular component (CC) annotation terms and biological process (BP) annotation terms. The length of column represent gene numbers enriched in this annotation term.

Comment 6: Important reviews about disease-miRNA association prediction and disease-lncRNA association prediction should be emphatically introduced (PMIDs: 29045685 and 27345524).

Response: Thanks for your valuable comment. We have discussed these computational models in the last paragraph, as follows:

Studies have demonstrated that identification of disease-associted miRNAs/lncRNAs has been will contribute to understand the molecular pathogenesis and to develop molecular tools for the prevention, diagnosis and treatment of diseases (73, 74). In this study, we identified several miRNAs and lnRNAs that significantly associated with alcohol-related EC by using correlation analysis with clinical characteristics and prognosis. In fact, many computational models were developed to identify disease-related miRNAs and lncRNAs (27-29). For example, Chen et al. developed a computational model of Matrix Decomposition and Heterogeneous Graph Inference for miRNA-disease association prediction (29) and developed the method of Laplacian Regularized Least Squares for LncRNA–Disease Association (LRLSLDA) in the semisupervised learning framework (28), these computational models have been confirmed to be effective. Additionally, computational models have been considered as important biological tool for biomedical research, which help to identify the most important miRNAs/lncRNAs-disease associations for experimental validation, markedly decreasing the cost and time for biological experiments (73, 74). Therefore, these computational models should be used for validation our findings and for identifying other potential disease-associated miRNAs and lncRNAs in the future.

Comment 7: The novelty of this work is not well highlighted. In addition, limitations of the work and work prospects were not provided.

Response: Thanks for your valuable comment. We have added the relevant content in the Discussion section, as follows:

In this paper, the molecular expression characteristics of alcoholic and non-alcoholic EC were systematically analyzed, and some lncRNAs, miRNAs and mRNAs that significantly correlated with clinical characteristics and overall survival of patients were screened out. In addition, the regulatory relationship among these key molecules was investigated. This study provides targets to investigate the pathological mechanism of alcoholic-related EC at molecular level. However, these investigations were concluded based on bioinformatics analysis and warrant further validation for clinical application via additional studies, like immunohistochemistry or quantitative real-time polymerase chain reaction.

Studies have demonstrated that identification of disease-associted miRNAs/lncRNAs has been will contribute to understand the molecular pathogenesis and to develop molecular tools for the prevention, diagnosis and treatment of diseases (73, 74). In this study, we identified several miRNAs and lnRNAs that significantly associated with alcohol-related EC by using correlation analysis with clinical characteristics and prognosis. In fact, many computational models were developed to identify disease-related miRNAs and lncRNAs (27-29). For example, Chen et al. developed a computational model of Matrix Decomposition and Heterogeneous Graph Inference for miRNA-disease association prediction (29) and developed the method of Laplacian Regularized Least Squares for LncRNA–Disease Association (LRLSLDA) in the semisupervised learning framework (28), these computational models have been confirmed to be effective. Additionally, computational models have been considered as important biological tool for biomedical research, which help to identify the most important miRNAs/lncRNAs-disease associations for experimental validation, markedly decreasing the cost and time for biological experiments (73, 74). Therefore, these computational models should be used for validation our findings and for identifying other potential disease-associated miRNAs and lncRNAs in the future. Similarly, we investigated lncRNA-miRNA interactions based on miRcode and starBase databases. While various network algorithms or models have been developed to predict lncRNA-miRNA interactions, which showed superior prediction performance than others (75, 77). In order to screen the most important lncRNA-miRNA interactions, different algorithms or methods should be used in the future. Moreover, circRNAs also play crucial roles in the occurrence and progression of many complex diseases, and several state-of-the-art computational models for predicting circRNA-disease associations have also been developed (78). Thus, investigation of circRNAs that associated with alcohol-related EC using these computational models should be the future direction of our work.

Comment 8: I want to know whether the framework proposed in this manuscript is effective for other cancers.

Response: Thanks for your valuable comment. The framework is also applicative for other cancers.

Thank you again for your valuable comments.

Reviewer #2

In the current manuscript, the authors explored the lncRNA–miRNA–mRNA networks involved in alcohol-related esophageal cancer (EC). They downloaded RNA sequencing and clinical data from TCGA. The lncRNA–miRNA–mRNA competing endogenous RNA (ceRNA) networks were constructed based on the information from miRcode, starBase, and miRTarBase databases. A total of 906 DEGs, 40 DELs, and 52 DEMs were identified. There were eight lncRNAs and eight miRNAs, especially ST7-AS2 and miR-1269, which were significantly associated with the survival rate of patients with EC. Besides, SNHG12–miR-1–ST6GAL1, SNHG3–miR-1–ST6GAL1, SPAG5-AS1–miR-133a–ST6GAL1, and SNHG12–hsa-miR-33a–ST6GA interactions, related to the N-Glycan biosynthesis pathway, may have key roles in alcohol-related EC. However, there are some problems to be further improved before acceptance for publication.

Comment 1: Literature review is somewhat incomplete in the introduction, especially about the progress of the bioinformatic analysis about the discovery of ncRNA related with EC.

Response: Thanks for your valuable comment. We have added the relevant content in the Introduction section, as follows:

Increasing number of studies has revealed the crucial roles of miRNAs/lncRNAs in the pathological progression of EC, including the proliferation, apoptosis, invasion, angiogenesis, metastasis, chemoradiotherapy resistance, as well as stemness of EC, indicating the potential clinical applications value of these non-coding RNAs as diagnostic and prognostic biomarkers (23-26). Many computational models were then developed to identify disease-related miRNAs and lncRNAs (27-29).

Comment 2: The flowchart in Fig.1 is too simple to show the process very well, please improve it. Also the authors should put Fig.1 before Fig.2.

Response: Thanks for your valuable comment. The Figure 1 has been revised, as follows:

All the figures are shown in order. Figure 1 is shown before Figure 2 in our manuscript.

Comment 3: In the Data acquisition section, the authors stated that they downloaded the sequencing data from the TCGA database. Is the authors sure about this? Usually we just download the Gene Expression Quantification data. If the authors processed the raw sequencing data, please write clearly the processing process.

Response: Thanks for your valuable comment. The RNA-seq fragments per kilobase of transcript per million fragments mapped (FPKM) data were downloaded from TCGA database.

Comment 4: Also about the Data acquisition. What is the “paired samples” mean? Were 18 normal samples used in the study? Please make these clearer.

Response: Thanks for your valuable comment. The RNA-seq (miRNAs and mRNAs) profile data of EC and corresponding clinical information were retrieved from TCGA database. A total of 198 mRNA and 198 miRNA sequencing sequenced data were obtained from IlluminaHiSeq_RNASeq and IlluminaHiSeq_miRNASeq sequencing platforms, respectively. The samples of mRNAs seq-data and the samples of miRNA seq-data were matched by the sequencing barcodes, and 190 samples were obtained after matching using sequencing barcodes, including 18 normal and 172 tumor samples. Among the 172 tumor samples, there were 120 alcohol-related EC tumor samples and 50 non-alcohol-related EC tumor samples according to the drinking status of the patients. The rest two samples had no records for drinking status. Finally, the 170 tumor samples were used in the following analysis.

Comment 5: Why is the criteria of DEG set to false discovery rate (FDR) < 0.05 and |fold change (FC)| > 1.5?

Response: Thanks for your valuable comment. The threshold |fold change (FC)| > 1.5 is equal to |log2 FC| > 0.585, which is a very common analytical threshold and has been used in many studies.

Comment 6: There are some misspellings of software names, such as “miRcorde”, “miRecorde”, “Cystoscope”. Please check these issues carefully.

Response: Thanks for your kindest reminding, and these misspellings have been corrected.

Comment 7: I suggest the authors should discuss more about the meaning of their research in the last section, as they use several online tools.

Response: Thanks for your valuable comment. We have added the relevant content according to your suggestion, as follows

In this paper, the molecular expression characteristics of alcoholic and non-alcoholic EC were systematically analyzed, and some lncRNAs, miRNAs and mRNAs that significantly correlated with clinical characteristics and overall survival of patients were screened out. In addition, the regulatory relationship among these key molecules was investigated. This study provides targets to investigate the pathological mechanism of alcoholic-related EC at molecular level. However, these investigations were concluded based on bioinformatics analysis and warrant further validation for clinical application via additional studies, like immunohistochemistry or quantitative real-time polymerase chain reaction.

Comment 8: Could the authors give some discussions whether their method and obtained result could be further used to predict EC-related ncRNAs? It could be considered as the future direction of their work. Some recommended studies are helpful (PMIDs: 33588070, 34232474, 34329377; DOI: 10.1016/j.knosys.2019.105261).

Response: Thanks for your valuable comment. We have discussed these computational models in the last paragraph, as follows (the underlined sentences):

Studies have demonstrated that identification of disease-associted miRNAs/lncRNAs has been will contribute to understand the molecular pathogenesis and to develop molecular tools for the prevention, diagnosis and treatment of diseases (73, 74). In this study, we identified several miRNAs and lnRNAs that significantly associated with alcohol-related EC by using correlation analysis with clinical characteristics and prognosis. In fact, many computational models were developed to identify disease-related miRNAs and lncRNAs (27-29). For example, Chen et al. developed a computational model of Matrix Decomposition and Heterogeneous Graph Inference for miRNA-disease association prediction (29) and developed the method of Laplacian Regularized Least Squares for LncRNA–Disease Association (LRLSLDA) in the semisupervised learning framework (28), these computational models have been confirmed to be effective. Additionally, computational models have been considered as important biological tool for biomedical research, which help to identify the most important miRNAs/lncRNAs-disease associations for experimental validation, markedly decreasing the cost and time for biological experiments (73, 74). Therefore, these computational models should be used for validation our findings and for identifying other potential disease-associated miRNAs and lncRNAs in the future. Similarly, we investigated lncRNA-miRNA interactions based on miRcode and starBase databases. While various network algorithms or models have been developed to predict lncRNA-miRNA interactions, which showed superior prediction performance than others (75-77). In order to screen the most important lncRNA-miRNA interactions, different algorithms or methods should be used in the future. Moreover, circRNAs also play crucial roles in the occurrence and progression of many complex diseases, and several state-of-the-art computational models for predicting circRNA-disease associations have also been developed (78). Thus, investigation of circRNAs that associated with alcohol-related EC using these computational models should be the future direction of our work.

Comment 9: English expressions need to be edited more careful and more native, in this manuscript, there are some mistakes.

Response: Thanks for your valuable comment. The mistakes have been checked and corrected in this revision. In addition, we have sent this manuscript to be edited by professional editors at Editage. The language editing certificate has been provided.

Thank you again for your valuable comments.

---

## [Decision Letter · Decision Letter 1]

29 Nov 2021

PONE-D-21-25185R1Construction of long non-coding RNA-and microRNA-mediated competing endogenous RNA networks in alcohol-related esophageal cancerPLOS ONE

Dear Mr. Yu Wang,

Thank you for submitting your manuscript to PLOS ONE. After careful consideration, we feel that it has merit but does not fully meet PLOS ONE’s publication criteria as it currently stands. Therefore, we invite you to submit a revised version of the manuscript that addresses the points raised during the review process.

We look forward to receiving your revised manuscript.

Kind regards,

Qi Zhao

Academic Editor

PLOS ONE

Journal Requirements:

Reviewers' comments:

Reviewer's Responses to Questions

**Comments to the Author**

1. If the authors have adequately addressed your comments raised in a previous round of review and you feel that this manuscript is now acceptable for publication, you may indicate that here to bypass the “Comments to the Author” section, enter your conflict of interest statement in the “Confidential to Editor” section, and submit your "Accept" recommendation.

Reviewer #1: All comments have been addressed

Reviewer #2: (No Response)

2. Is the manuscript technically sound, and do the data support the conclusions?

Reviewer #1: Yes

Reviewer #2: Yes

3. Has the statistical analysis been performed appropriately and rigorously? 

Reviewer #1: Yes

Reviewer #2: Yes

4. Have the authors made all data underlying the findings in their manuscript fully available?

Reviewer #1: Yes

Reviewer #2: Yes

5. Is the manuscript presented in an intelligible fashion and written in standard English?

Reviewer #1: Yes

Reviewer #2: Yes

6. Review Comments to the Author

Reviewer #1: The authors have addressed all my comments. I think the revised manuscript is acceptable for publication.

Reviewer #2: There are many mistakes in the information of references. The authors should carefully check and revise them. For example, the reference [76] has no journal name and the author’s information is incomplete; the reference information of [77] should be 2020, 191: 105261, and the reference information of [78] should be 2021, 22(6): 1-27 with DOI: 10.1093/bib/bbab286.

7. PLOS authors have the option to publish the peer review history of their article (what does this mean?). If published, this will include your full peer review and any attached files.

Reviewer #1: **Yes: **Nana Guan

Reviewer #2: No

---

## [Author Response · Author response to Decision Letter 1]

6 Dec 2021

Dear editor,

We would like to extend our most sincere gratitude to the reviewers' comments on our manuscript entitled "Construction of long non-coding RNA-and microRNA-mediated competing endogenous RNA networks in alcohol-related esophageal cancer ". Those comments are all valuable and very helpful for revising and improving our paper, as well as the essential guiding significance to our researches. We have revised the comments extensively and have made the corrections in order to meet with approval. The leading corrections in the paper and the response to the reviewers' comments are as follows.

We appreciate for Editors/Reviewers' warm work earnestly and hope that the correction will meet with approval.

Once again, thanks for the comments and suggestions.

Sincerely,

Quan Du

A point-by-point response to editor’s and reviewer’s comments

Academic Editor

Journal Requirements:

Response: Thank you for your valuable comment. We have checked all the references, and no retracted papers are cited in our manuscript.

In this revision, no references are deleted, and doi number are added for references if their doi number are available.

Reviewer #1

The authors have addressed all my comments. I think the revised manuscript is acceptable for publication.

Response: Thank you for your recommendation for our manuscript.

Reviewer #2

There are many mistakes in the information of references. The authors should carefully check and revise them. For example, the reference [76] has no journal name and the author’s information is incomplete; the reference information of [77] should be 2020, 191: 105261, and the reference information of [78] should be 2021, 22(6): 1-27 with DOI: 10.1093/bib/bbab286

Response: Thank you for your valuable comment. We have checked all the references to correct the mistakes.

The revised references 76-77 are as follows:

76. Zhang L, Yang P, Feng H, Zhao Q and Liu H: Using Network Distance Analysis to Predict lncRNA-miRNA Interactions. Interdiscip Sci. 2021. 13 (3):535-545. doi:10.1007/s12539-021-00458-z

77. Liu H, Ren G, Chen H, Liu Q and Zhao Q: Predicting lncRNA-miRNA interactions based on logistic matrix factorization with neighborhood regularized. Knowledge-Based Systems. 2020. 191:105261. doi:10.1016/j.knosys.2019.105261

78. Wang CC, Han CD, Zhao Q and Chen X: Circular RNAs and complex diseases: from experimental results to computational models. Briefings in bioinformatics. 2021. 22 (6):1-27. doi:10.1093/bib/bbab286

---

## [Decision Letter · Decision Letter 2]

27 May 2022

Construction of long non-coding RNA-and microRNA-mediated competing endogenous RNA networks in alcohol-related esophageal cancer

PONE-D-21-25185R2

Dear Dr. Wang,

We’re pleased to inform you that your manuscript has been judged scientifically suitable for publication and will be formally accepted for publication once it meets all outstanding technical requirements.

Kind regards,

Yun Zheng, Ph.D

Academic Editor

PLOS ONE

Additional Editor Comments (optional):

Reviewers' comments:

Reviewer's Responses to Questions

**Comments to the Author**

1. If the authors have adequately addressed your comments raised in a previous round of review and you feel that this manuscript is now acceptable for publication, you may indicate that here to bypass the “Comments to the Author” section, enter your conflict of interest statement in the “Confidential to Editor” section, and submit your "Accept" recommendation.

Reviewer #2: All comments have been addressed

2. Is the manuscript technically sound, and do the data support the conclusions?

Reviewer #2: Yes

3. Has the statistical analysis been performed appropriately and rigorously? 

Reviewer #2: Yes

4. Have the authors made all data underlying the findings in their manuscript fully available?

Reviewer #2: Yes

5. Is the manuscript presented in an intelligible fashion and written in standard English?

Reviewer #2: Yes

6. Review Comments to the Author

Reviewer #2: All of my comments have been addressed. I think the revised manuscript is acceptable for publication.

7. PLOS authors have the option to publish the peer review history of their article (what does this mean?). If published, this will include your full peer review and any attached files.

Reviewer #2: No

---

## [Editor Report · Acceptance letter]

6 Jun 2022

PONE-D-21-25185R2 

Construction of long non-coding RNA- and microRNA-mediated competing endogenous RNA networks in alcohol-related esophageal cancer 

Dear Dr. Wang:

I'm pleased to inform you that your manuscript has been deemed suitable for publication in PLOS ONE. Congratulations! Your manuscript is now with our production department. 

Kind regards, 

on behalf of

Dr. Yun Zheng 

Academic Editor

PLOS ONE